# Cold Tolerance Mechanisms in Mungbean (*Vigna radiata* L.) Genotypes during Germination

Lekshmi S. Manasa [1,2], Madhusmita Panigrahy [1], Kishore Chandra Panigrahi [1], Gayatri Mishra [1,3], Sanjib Kumar Panda [4,*] and Gyana Ranjan Rout [2,*]

1   National Institute of Science, Education and Research, Homi Bhabha National Institute (HBNI), Jatni 752050, India
2   Department of Agricultural Biotechnology, Odisha University of Agriculture & Technology, Bhubaneswar 751003, India
3   Department of Biology, University of Utah, Salt Lake City, UT 84112, USA
4   Department of Biochemistry, Central University of Rajasthan, Ajmer 305817, India
*   Correspondence: sanjib.panda@curaj.ac.in (S.K.P.); grrout@rediffmail.com (G.R.R.)

**Abstract:** Mungbean or greengram (*Vigna radiata*) is an important legume crop well known for its high protein with nitrogen-fixing abilities. However, the severe yield loss in mungbean occurs due to susceptibility to low temperatures at all stages of plant growth including germination and is a serious concern for its cultivation and productivity. To select cold-tolerant genotypes, a germination-based screening at 10 °C was performed in a total of 204 germplasms. The study showed that cold stress of the initial 8-days during seedling establishment imposed a negative impact throughout the life of mungbean genotypes, which were reflected in the vegetative and reproductive phase (plant height, days to 50% flowering and pods/plant, seeds/pod, yield/plant, and 100-seed weight). The biplot analysis showed that parameters such as germination rate index, Timson's index, mean germination time, and coefficient of the velocity of germination are the key influential germination parameters for identifying cold tolerance in the seedling stage. Identified cold-tolerant genotype (PAU911) retained higher rootlet number, leaf area, and increased chlorophyll, carotenoid, and malondialdehyde (MDA) content at 10 °C. Based on the confocal microscopic study, it is noticed that the stomatal density, open pore percentage, and trichome density were significant differences in seedlings exposed to cold stress as compared to non-stress. On the basis of matrix-assisted laser desorption ionization-time-of-flight mass spectrometry (MALDI-TOF MS) analysis, it is observed that a new protein identified as *TETRATRICOPEPTIDE-REPEAT THIOREDOXIN-LIKE1* (*TTL1*) (UNIPROT Identifier: LOC106762419) which highly correlated with the cold stress response of in the cold-tolerant genotype. Our study identifies a noble member, TTL1, whose expression has a positive role in cold tolerance response at the protein level in *V. radiata*. This study will help breeding programs with regard to the sustainable growth of mungbean.

**Keywords:** cold stress; greengram; mungbean; seed germination; screening; *Vigna radiata*; TTL1 gene expression

## 1. Introduction

A serious threat is posed to agricultural productivity due to the deterioration in crop yield caused by various environmental factors, diseases, and pathogen attacks along with climate change and unpredictable weather conditions of concern [1]. Moreover, world crop production needs to be enhanced by ~50% in 2050 in order to fulfill the nutrient requirement of ~10 billion people [1]. Plant biologists and breeders are facing a great challenge in order to improve crop yield along with developing stress-tolerant crops. In particular, cold stress is a serious threat to sustainable crop yield [2,3]. Cold stress is categorized into chilling stress (0 °C to 20 °C) or freezing stress (below 0 °C) according to the category of adverse effects of damages they cause [4,5]. With very few exceptions, the genotypes of the tropical

region are sensitive to chilling temperatures (below 0 °C) during almost all stages of plant development, including seed germination, vegetative growth, and the reproductive phase. Cold stress (including both chilling and freezing) is, therefore, one of the major stresses that significantly dampen crop production [6]. The cold signal is perceived by receptors of the cell membrane, the severity of which is determined by factors such as light and temperature [4]. The signal is transduced by cold-responsive genes [6,7], second messengers such as calcium, phosphatidic acid [2,8,9], abscisic acid, and transcription factors [10–12] to exert symptoms including decreased germination, stunted growth, chlorosis, wilting of leaves, compromised reproductive development and ultimately necrosis [3,13,14].

Chilling stress induces irreversible electrolyte leakage from the cell, reduction in scavenging enzymes' activity, loss of energy in photosynthetic apparatus, and stability of proteins [2]. Cold tolerance is induced by limiting the membrane fluidity, increasing the membrane rigidity, probably by alteration of cytoskeleton organization, and calcium flux. These are achieved by the accumulation of osmolytes such as proline [8], malonaldehyde [15,16], dehydrins [17], increasing abundance of highly hydrophobic proteins such as cold shock proteins, late embryogenesis abundant proteins (LEA), anti-freezing proteins [2]. At the molecular level, cold acclimation is associated with the activation of MAPK cascades, up-regulation of a cold-related gene (COR), Inducer of CBF Expression1 (ICE1), and expression of CRT binding factor (CBF)/DRE binding (DREB) proteins [17]. Cold-tolerant cultivars grow vigorously at seedling stages, become established faster than cold-sensitive ones even under cold stress, and bear higher adaptability with yields of high-quality fruit when grown under sub-optimal temperatures [16]. Moreover, cold tolerant cultivars could be planted early and harvested early when the crop may have higher values [18].

Mungbean (Greengram), *Vigna radiata*, a tropical pulse crop largely grown under semi-arid and subtropical environments, is shown to have sensitivity to low-temperature regimes [19]. Though mungbean has a multifarious advantage, sensitivity to chilling stress can largely limit its productivity above the threshold mark [20]. Low temperatures can hamper the germination of the seeds [21] and delay plant growth and development, leading to variability in crop maturation and the reproductive ability of plants [22]. During later stages, plant growth is highly retarded, leading to reduced flowers or fruit. Furthermore, low soil temperatures below 15 °C, may often restrict direct seeding [23–26], highlighting the importance of edaphic factors accountability in plant establishment. Scanty studies are reported for screening, identifying chilling tolerant g1enotypes as well as for determination of the characteristics at the seedlings stage that can be taken as a marker for selecting cold-tolerant genotype [27]. The severe susceptibility of mungbean to chilling stress and also lack of suitable resistant genotypes have thus become the reason for the decrease in mungbean productivity [28]. Therefore, identifying and developing mung genotypes that are tolerant to cold stress is the need for the hour. The present study is aimed at the screening of mungbean genotypes for cold stress response at 10 °C during the germination stage and the selection of cold tolerant and susceptible genotypes. It also includes a study of the phenotypic traits and physiological, biochemical, and anatomical parameters contributing to the cold tolerance of the representative set of tolerant and susceptible genotypes. The primary step includes screening the genotypes at the very germination stage, keeping in mind the ability of the seeds to establish under cold conditions.

## 2. Materials and Methods

### 2.1. Plant Material and Growth Condition

A total of 204 Vigna radiata accessions that include 127 Indian varieties, 58 Odisha landraces, 15 recombinant Inbred lines (RIL) along with 4 high-yielding released varieties (OBGG-52, KAMDEV, IPM-02-3, IPM-02-14) were used in this study (Supplementary Table S1). The four high-yielding varieties abbreviated as C1, C2, C3, and C4 respectively were taken as checks against the 200 lines in test. All the accessions were obtained from field-grown conditions (in polybags pots 1:1; loam and cow dung) with augmented design during August 2019 at research station of the Odisha University of Agriculture and Technology,

Bhubaneswar, India with geographic details of 20.2650° N, 85.8117° E, 25.9 m above the sea level. The average day/night temperature was 30.5 °C/25.5 °C and the relative humidity was 86% within average of ~8.1 h (h) average day length. Irrigation was conducted for 3 days every 7-dayinterval.

### 2.2. Screening at Seedling Stage

For screening under cold stress, the *V. radiata* seeds were soaked in water for 6 h followed by germination induction under white light (WL) for 6 h. The seeds were sown uniformly at 1.5 cm depth in pro-trays containing soil mixture as used by [13]. Germination assay was conducted for 8 days followed by recovery assay of 2 days in plant growth chambers (Model No. CU36L6, Percival USA) maintained separately either at 10 °C (cold stress: CS) or 22 °C (control condition: CC) with 70% relative humidity at National Institute of Science Education and Research (NISER), India. Light was obtained from Philips 17 W F17T8/TL741 USA Alto II technology tubes with 40% light intensity, which is equivalent to ~67.5 μmol m$^{-2}$ s$^{-1}$. The growth chambers were programmed to provide cycles of 18 h light and 6 h dark (Short Day: SD). All seedling experiments were performed in 3replicationswith 10 seedlings from each accession. Data presented is a mean of 2 biological replicates. Scoring for germination was conducted on the 2nd, 4th, 6th, and 8th days (d) after sowing (DAS). Radicle emergence of at least 2 mm was counted to be germinated. On 8th day the plants were shifted from 10 °C to 22 °C chamber and grown for 2 d for acclimatization. Further, all the plants were shifted to net houses for growth under natural day-night conditions.

### 2.3. Phenotypic Assessment for Cold Stress Tolerance in Vegetative and Reproductive Stage

Several phenotypic parameters including germination %germination (GP), 50% emergence time (50E), seedling height on 8th day, plant height (PH), days to 50% flowering (DFF), number of pods/plants, seeds/pod, test weight, and yield/plant were taken in the net house at NISER under natural day-night conditions. Germination parameters including seedling length (SL), seedling vigor (SV), germination rate index (GRI), mean germination time (MGT), coefficient of velocity of germination (CVG), mean germination rate (MGR) [29] and Timson's germination index (TI) Khan and Unger, 1998 [30], 50% emergence (50E) were computed from the data obtained till 8th DAS. The leaf area (LA) was taken using ImageJ software (version No: 1.53k) (Supplementary Table S2).

Relevant stress indices, such as Promptness Index (PI) [31], Germination stress index (GSTI), Plant Height stress index (PHSI) [32], Stress Susceptibility Index (SSI) [33], Tolerance Index (TOL) [34], Mean Productivity Index (MPI) [34], Stress Tolerance Index (STI), Geometric Mean Productivity (GMP) [35], Resilience Capacity index (RCI) and Productive Capacity Index (PCI) was also computed by using the formula mentioned in Supplementary Table S2. Each of the stress parameters is the mean of 15 observations from either control or cold condition.

Seedling phenotype such as seedling root length (RL), seedling length (SL), rootlet number (RLN), leaf area (LA) was noted from the 3 selected lines and each data was a mean of 30 observations from 3 biological replicates.

### 2.4. Statistical Analysis

For the dendrogram, the cold stress indices of the 204 accessions were subjected to clustering analysis based on the hierarchical clustering approach [36]. The 'hclust' function for the hierarchical clustering of the genotypes in Rstudio (Version No: 2021.09.0). Cluster Analysis and Dendrogram were conducted by grouping the germplasms, and a dendrogram was generated based on the selected stress tolerance indices.

Correlation analysis was made using 'corrplot' function among the parameters and represented using a corelogram using Rstudio (Version No 2021.09.0).

Inter-clustral and Intra-clustral analyses were conducted for parameters such as germination percentage (GP), germination stress index (GSTI), seedling height stress index

(SHSI), resilience capacity index (RSI), and productive capacity index (PCI). Differences in the above-mentioned 5 parameters were shown using Violin plots.

Principal Component Analysis (PCA) analysis was computed for 16 parameters among 204 genotypes using a 'prcomp' function from Factoextra library package in Rstudio (Version No:2021.09.0) and a biplot was conducted using the 'ggplot' function. The eigen values generated from the PCA analysis were presented in the screeplots (Supplementary Figure S1).

*2.5. Physiological and Biochemical Analysis*

Three genotypes (G77-PAU911, G88-PUSA 1672, and G92-PUSA 9531) were selected based on stress screening of 204 genotypes for physiological, anatomical, and biochemical analysis. Physiological parameter such as pigment content (chlorophyll and carotenoid), relative water content, and biochemical parameters such as malondialdehyde (MDA), total sugar content, superoxide dismutase (SOD) enzyme activity, and reactive oxygen species (ROS), were estimated as explained below from the 3 selected lines and each data was a mean of 30 observations from 3 biological replicates. For all these measurements, statistical significance among the genotypes at 10 °C and 22 °C were calculated using Graphpad Prism 8.0.1.

Pigment content was estimated from ~25 mg of leaf sample (from two-leaf seedlings samples on the 8th DAS was frozen in liquid nitrogen and extraction of total chlorophyll and carotenoid was conducted in 3 mL80% acetone extracts according to [37] with small modification [38]. Estimation of chlorophyll and carotenoids were conducted by measurement of absorbance (A) at 645 nm 663 nm and 480 nm by using the following formulas:

$$\text{Carotenoid } (\mu g/mg) = ((A480 + (0.114 \times A663) - (0.638 \times A647)) \times 3000)/(1000 \times FW)$$

$$\text{Chlorophyll } (\mu g/mg) = 20.2 \, (A645) + 8.02 \, (A663)$$

For Relative water content (RWC), 1st and 2nd leaf of each genotype on the 8th DAS, at two-leaf stage, was cut and weighed for fresh weight. They were then floated in 50 mL falcons for 2 days at RT for turgid weight. They were dried completely till the weight was constant at 70 °C for 2–3 days.

$$\text{RWC} = ((FW - DW)/(TW - DW)) \times 100$$

where: FW = Fresh weight; TW = turgid weight.

For malondialdehyde (MDA) assay, ~100 mg of the leaf samples (8-d-old seedlings) were ground in 0.25% of Tertiary butyl alcohol (TBA) and 10% trichloroacetic acid (TCA). Further, it was incubated at 95 °C for 30 min, centrifuged at 1000 rpm for 10 min. The absorbance of the supernatant was determined spectrophotometrically at 532 nm and 600 nm.

Total sugar content was estimated as explained by Buysse et al. [39] and with small modifications [40]. Briefly, ~50 mg fresh weight of leaves was ground with 5 mL of 80% methanol and the ground extract was boiled at 95 °C for 5 min. The supernatant was collected in a test tube. Further, 5 mL of 80% (*v/v*) methanol was added and boiled for 2 min. The supernatant of the two steps was added and measured. From the supernatant, 1 mL was taken out in another test tube. Then 1 mL of 18% phenol and 5 mL of conc. $H_2SO_4$ was added to the test tube. It was mixed well in a vortex mixture for 20 min. Absorbance was taken at 490 nm before 1 h of mixing. The sugar estimation was conducted using glucose standard prepared from a mixture of Glucose: Fructose: Galactose at 1:1:1 ratio. Proline content was estimated from ~20 mg of leaf sample (from two-leaf seedlings samples on the 8th DAS) was frozen in liquid nitrogen and ground in 1000 μL of 70% (*v/v*) alcohol. Further, 1000 μL of reaction mixture (Ninhydrin (1%), Acetic acid (60%), ethanol (20%) in distilled water), and 500 μL of leaf extract were taken and mixed. The mixture was heated at 95 °C for 20 min and further centrifuged at 10,000 rpm for 1 min and the supernatant was

measured spectrophotometrically at 520 nm. The proline was estimated from the standard curve prepared from the different dilutions of L-proline. The amount of proline in each sample was estimated using the formula

Proline in nmol mg$^{-1}$ fresh weight (FW) = [(Abs extract blank)/slope] × [Vol extract/Vol aliquot × [1/FW].

For membrane stability study, about 20 pieces of leaves were cut into small pieces and kept in 20 mL of autoclaved mili Q water overnight and electrical conductivity was measured after 24 h. The samples were then autoclaved and electrical conductivity was measured. Parameters such as membrane leakage and Injury index were calculated using the formula

Electrolyte Leakage = [(Ecf − Eci)/(Ect − Eci)] × 100

Index of Injury = [(RS − RC)/(1 − RC)] × 100

where, Eci = Electrical conductivity before treatment; Ecf = Electrical conductivity after treatment; Ect = Electrical conductivity after autoclave.

RS = (Ecf − Eci)/(Ect − Eci) for Control condition,

RC = (Ecf − Eci)/(Ecf − Eci) for Cold stress

Superoxide dismutase (SOD) enzyme activity was determined spectrophotometrically as described by Dhindsa et al. [41] with small modifications [42]. For these assays, 100 mg of leaf sample (1st and 2nd leaf 8-d-old seedling) was ground with 4 mL of extraction buffer (0.1 M phosphate buffer, pH 7.5, containing 0.5 mM EDTA) and filtered through 4 layers of cheesecloth. The filtrate was transferred to centrifuge tubes and centrifuged at 15,000 rpm for 20 min. The supernatant was used as the enzyme extract and 100 μL of this extract was used for each enzyme activity assay. Briefly, SOD activity was determined by measuring the decrease in the absorbance of blue-colored formazone and O$^{2-}$ at 560 nm.

For reactive oxygen species (ROS) estimation, nitrobluetetrazolium (NBT) staining was used. Notably, 10 mL of the NBT staining solution (0.2% NBT in 10 mM Phosphate buffer at pH 7) was added to each of the leaves (8-d-old). Vacuum was provided 3 times at 5 min intervals using vacuum desiccators and left overnight. NBT solution was removed and 15 mL compound solution (Acetic acid: glycerol: ethanol at 1:1:3) was added and kept at 95 °C for 15 min. After cooling, it was stored in storage solution (50 mL glycerol in 200 mL ethanol). Amount of ROS was estimated by calculating the percentage of stained area to the total area using ImageJ software version 1.53 k.

*2.6. Anatomical Analysis*

Anatomical analysis was performed by using 3 selected genotypes. All the measurement was made from the mean of 30 observations with 3 biological replicates. For all these measurements, statistical significance among the genotypes grown at 10 °C (Cold stress: CS) and 22 °C (Control condition: CC) was calculated using GraphPad Prism 8.0.1. Parameters such as stomatal density and open pore % trichome density were analyzed.

Freshly harvested tissues were cut into <1 cm of length and breadth using a razor blade. The samples were fixed either in cytoskeleton buffer or (Formalin-Acetic Acid-Alcohol) FAA. The fixed tissues were vacuumed for 15 min at an interval of 5 min interval for 4 times. Further, it was kept overnight under vacuum until the tissue sinks to the bottom. The FAA fixed tissues were stained with 0.1% acridine orange and were incubated for 1 h. The tissue was then mounted on a slide with a drop of 100% (*v/v*) glycerol. The cover slip was sealed with transparent sealer nail polish and was kept at 4 °C until it was imaged under a confocal microscope (Leica SP8; Leica Application Suite X3.5.5.19976) for trichome density. The quantification was conducted using Fiji ImageJ software.

In another set of experiments, FAA fixed tissues were stained with 1:1000 dilution DAPI in $1 \times$ PBS and incubated for 1 h. The tissue was then mounted on a slide with a drop of 100% glycerol. The cover slip was sealed with transparent sealer nail polish and was kept at 4 °C until it was imaged under a confocal microscope for stomatal pore open percentage and stomatal density. The quantification was conducted using Fiji ImageJ software.

Third set of experiment, the FAA fixed tissue was then mounted on a slide with a drop of 100% (*v/v*) glycerol. The cover slip was sealed with transparent sealer nail polish and was kept at 4 °C until it was imaged under a confocal microscope for lambda scanning [43]. The acquisition was in XYZ format with a zoom factor of 0.75. The Line average was 4 with a format of $512 \times 512$ with a speed of 400. The autofluorescence detection was at 488 nm laser with a laser power of 34% for Trichomes and 12.3% for others. The band width was in the range of 500–700 with 55 steps. The lambda detection step size was at 3.52 nm and the detection band width was 10. Using lambda scan, mesophyll cells per unit area were imaged. The quantification was conducted using Fiji ImageJ software.

## 2.7. Protein Extraction, SDS-PAGE

SDS-Poly Acrylamide Gel Electrophoresis (PAGE) was conducted according to Panigrahy [42]. Leaf samples (~0.5 g) of three genotypes and the control (grown at either at 10 °C or 22 °C) were homogenized with 2 mL of protein extraction buffer with following composition.

Protein Extraction Buffer:

| | |
|---|---|
| Tris-HCl (pH 6.8) | 50 mM |
| Tris-Glycine (pH 8.3) | 50 mM |
| Ethylenediamine-tetraaceticacid (EDTA) | 50 mM |
| β-Mercaptoethanol | 0.1% |
| Sucrose | 0.5 M |
| KCl | 0.1 M |
| PhenylmethylsulfonylFluoride (PMSF) | 2 mM |
| SDS (Sodium dodecyl sulphate) | 2% |

The protein concentrations in the samples were estimated following amido black method [42]. Briefly, 10 µL of protein sample with 250 µL of amido black solution is vortexed for 10 min and centrifuged for 10 min at highest speed. The pellet obtained was mixed with 250 µL of washing solution (10% acetic acid + 90% methanol) and briefly vortexed, centrifuged for 5 min at maximum speed. The washing step was repeated with pellet till it was colorless. Finally, the pellet was dried and resuspended in 250 µL 0.2 M sodium hydroxide. The OD was measured at 595 nm. For electrophoresis, resolving gel (10%) contained 390 mM Tris-HCl, pH 8.8, 10% (*w/v*) SDS, 10% (*w/v*) ammonium persulfate, and 0.4 µL ml$^{-1}$ TEMED and the stacking gel (5%) contained 130 mM Tris-HCl, pH 6.8, 10% (*w/v*) SDS, 10% (*w/v*) ammonium persulfate and 0.2 µL mL$^{-1}$ TEMED. Nearly 30 µg of protein sample was loaded in each well.

## 2.8. MS and Bioinformatics Analysis

The gel bands were distained and dissolved in 50% acetonitrile (ACN), dehydrated in 300 µL ACN, and rehydrated with 100 µL of 10 mM 1,4-Dithiothreitol (DTT)at 56 °C. After incubation DTT solution was removed the gel pieces were incubated with 100 µL of 55 mM Iodoacetamide in dark for 45 min. The supernatant was removed and the gel was incubated with ammonium bicarbonate solution for 10 min. Added 0.1 µg of Trypsin solution and soaked overnight at 37 °C on the thermomixer at 300 rpm. The gel pieces were extracted with 0.1% Trifluoroacetic acid (TFA) and dried completely. The dried peptide mix was suspended in (50/50 Ceric ammonium nitrate (CAN):Water + 0.1%TFA) and mixed with a saturated α-cyano-4-hydroxy cinnamic acid (HCCA) matrix in 1:1 ratio and the resulting 2 µL was spotted onto the MALDI Anchor plate. MALDI TOF/TOF MSDaltonicsUltraflex III (BRUKERS) having a laser beam was used to obtain the Peptide Mass Fingerprint of

the protein. Flex Control Software was used obtain a good intensity spectrum with a mass range of 500 to 5000 $m/z$. Spectra was then exported on to the software Biotools and searched on MASCOT database. Mungbean proteins fasta files were downloaded from NCBI and prepared in a database in Mascot [Matrix Science] licensed software for the analysis. MASCOT search parameters were: Fixed modification: Carbamidomethyl @ Cysteine, Variable modification: Oxidation @ Methionine, Enzyme: Trypsin (As the same enzyme used for digestion) and Missed Cleavage: 2. All MS and bioinformatics analyses were performed through Bruker Autoflex II Maldi MS (Germany) Novelgene Technologies Pvt. Ltd. Hyderabad.

## 3. Results

### 3.1. Effect of Cold Stress on Germination and Recovery at Seedling Stage

The effect of cold stress (CSat 10 °C) on germination in *V. radiata* seedlings was studied till the 8th day and genotypes were grouped according to the percentage of germination under CS (Figure 1). Germination at the control condition (CCat 22 °C) was 100% which indicated the status of healthy and viable seeds. Under CS, there was a significant ($p < 0.0001$) decrease in germination % median value (0.6) which was in the range of 20% to 70% compared to control when considering the inter-quartile range (IQR) (Figure 1A). Germination Index (GI) interrelates the germination percentage with time. GI showed a nearly5-fold decrease (from 60 to 12.5 median value) under CS as compared to the CC (Figure 1B) indicating a severe curb on growth and establishment of the germinated seed under CS. The seedling length (SL) of the seedlings after 8days (Figure 1C) also showed a significant decrease (~7.1 fold) under CS when compared with that of CC. As, the median SL in the CC-grown seedlings (i.e., 12.2 cm) was lesser than that of the CS-grown seedlings (i.e., 1.7 cm). These results indicated that CS dampened the growth and establishment of the seedlings more harshly than the germination. Population distribution according to germination percentage (Figure 1D) and 100% recovery (Figure 1E) was studied under CS to acquire an overall impression of resilience under cold for the genotypes in the study. The check genotype IPM-02-14 fell into the higher germination percentage range of 81–90% (Figure 1D) and could recover in the range of 90–100% at CS (Figure 1E). The graphs showed that the maximum number of genotypes (i.e., 15%) fell in the interval of 61–70% germination. Lines with higher germination % (i.e., 71–80%, 81–90%, and 91–100%) could also recover at a higher %,as these lines had recovery above 60% (Figure 1E). Lines with 41–50% and 61–70% germination under CS included the cases where >60% recovery was 46% and 29% respectively. These results indicated that germination under CS correlated with % recovery under CC after a CS.

### 3.2. Effect of Cold Stress Germination on Growth at Vegetative and Reproductive Stage

Germination and initial seedling establishment under CS for 8 d impacted significantly the further growth of the plant at vegetative and also at reproductive stages including several yield parameters, when grown in natural day-night conditions for 25–30 d (Figure 2). The median for plant height of CC and CS is 34.83 cm and 29.75 cm respectively (Figure 2A). Hence, there was ~20% decrease in plant height in the plants that were CS in the early seedling stage. Days to 50% flowering (DFF) (Figure 2B) showed a significant difference even though it did not follow a particular trend. The majority of plants that experienced CS in the early seedling stage flowered earlier by ~5 $\pm$ 5 days than the control. In the case of pods/plant (Figure 2C) there was a significant decrease of 54% in the case of cold-stressed plants. Seeds/pod (Figure 2D) surprisingly showed an increase of 14% in the plants that were CS in the early seedling stage when compared to the control population, although it was not significantly different between CC and CS. Considering yield per plant (Figure 2E), the median values for control and CS plants were 1.124 g and 0.469 g, respectively indicating a 58.2% decrease in yield/plant in the plants with CS in the early seedling stage. The median for 100-seed weight was 4.8 g for the control population and 3.6 g for the test population respectively indicating a 33.3% decrease in the CS plants.

These results indicated that though the CS was imposed at the germination and initial seedling establishment stage, it could significantly affect the morphological parameters at the vegetative and reproductive stages including some yield parameters.

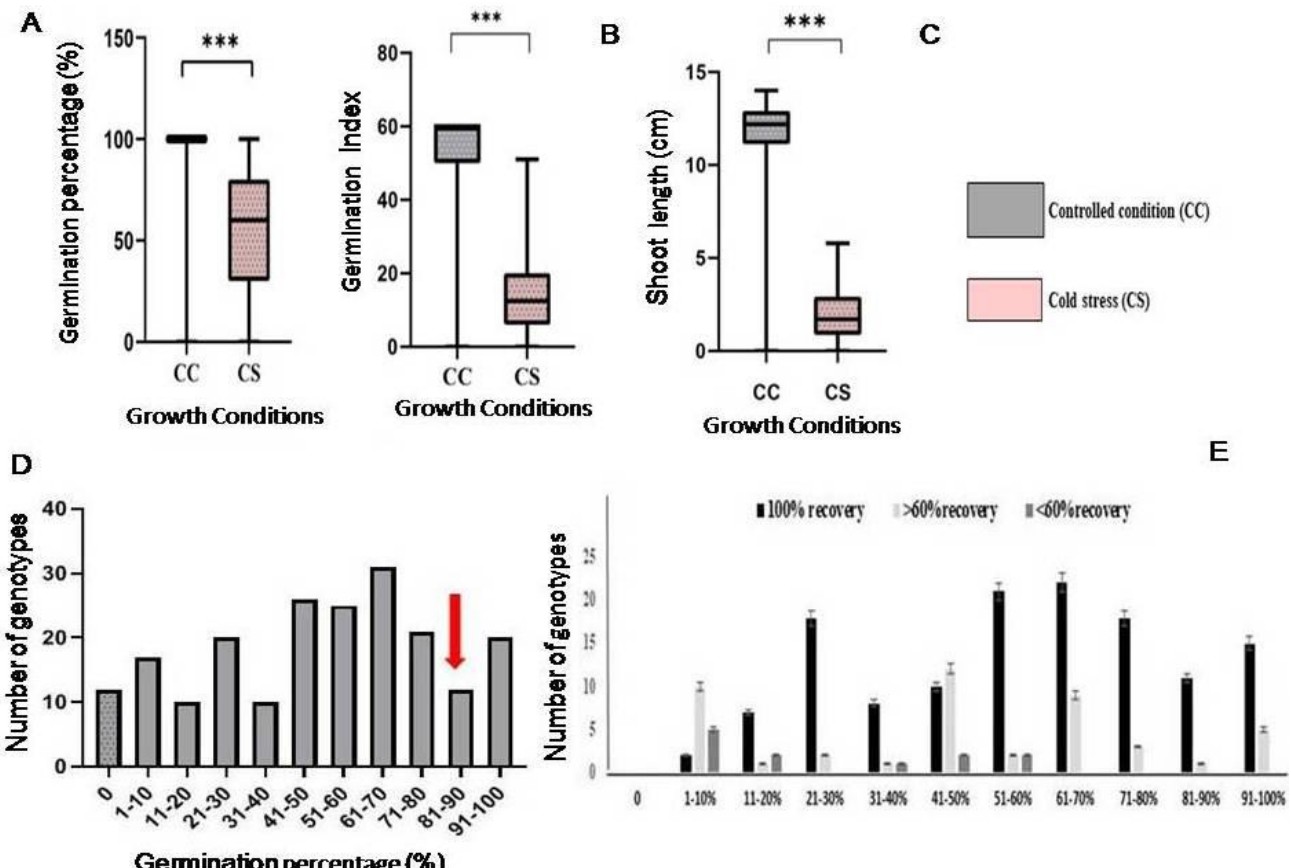

**Figure 1.** Germination percentage and recovery of *V. radiata* genotypes. Notably, 204 *V. radiata* genotypes were phenotyped for germination % at 10 °C on 8th day after sowing (DAS). (**A**) Germination %, (**B**) Germination Index, (**C**) seedling shoot length at 22 °C (control, CC) or 10 °C (Cold stress, CS), (**D**) germination % of population categorized into intervals of 10% each. RED arrow indicates well known high yielding genotype IPM-02-14. Colored arrows indicate the germination % of G88, G77 and G92, mentioned in the Section 3.6 of the text. (**E**) This figure shows genotypes with 100%, >60% and <60% recovery rate. Recovery was assayed by keeping the plants for recovery at 22 °C for 7 more days under CC after 8 days CS. Recovery % was calculated from the ratio of number of plants that survived to the total number of plants after CS.RED arrow indicates recovery of genotype IPM-02-14. Significance was conducted using unpaired *t*-test and is indicated as ***, which indicates $p < 0.005$.

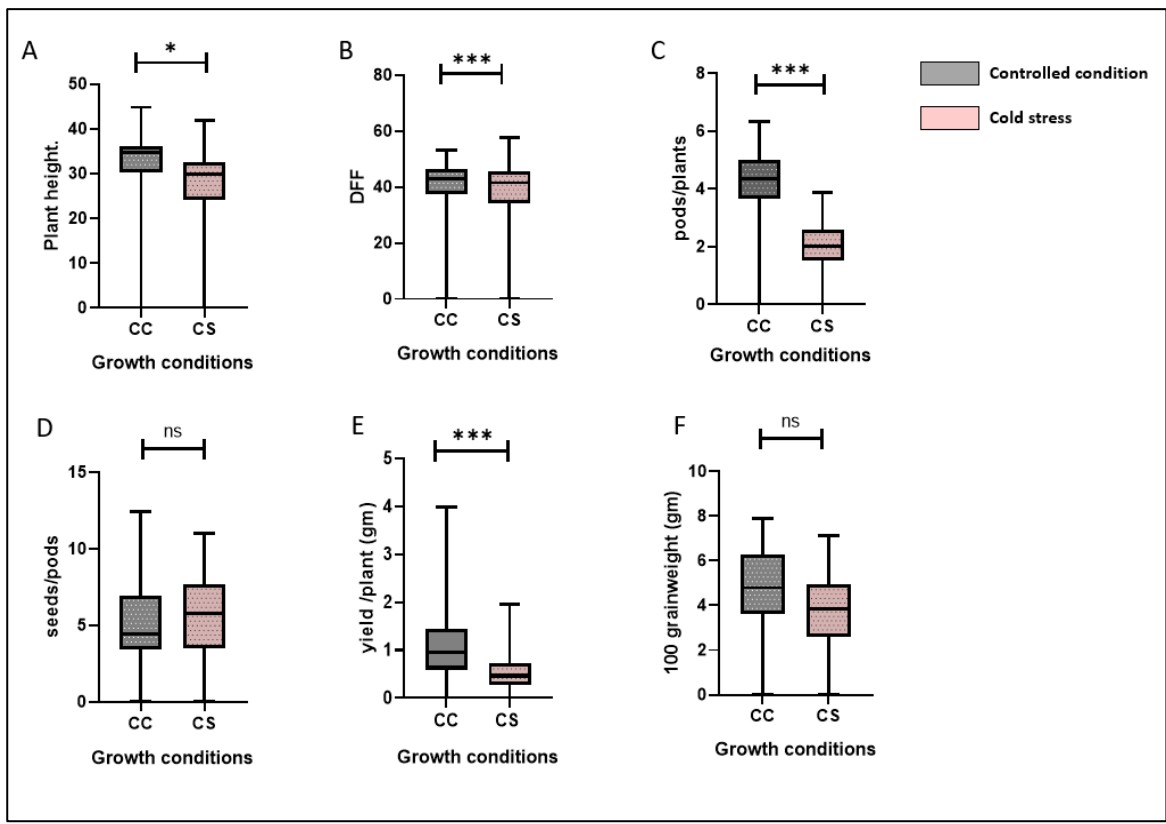

**Figure 2.** Effect of CS germination on growth at vegetative, reproductive stages and yield parameters. (**A**) Plant height (PH), (**B**) Days to 50% flowering (DFF), (**C**) Pods/plant, (**D**) Seeds/pods, (**E**) Yield/plant and (**F**) 100 Grain weight. Seeds were germinated either at 22 °C or 10 °C for 8 days after sowing, followed by growth at natural day-night conditions for 25–30 days. Data phenotyped at 25–30 DAS of vegetative stage, 56–75 DAS of reproductive stage, and post-harvest from 15 plants in 3 biological replicates. Data represents the median with mean indicated by line in the boxes. Significance was conducted using unpaired *t*-test and is indicated as * is $p < 0.05$; ***, $p < 0.005$.

### 3.3. Correlation among Phenotypic Traits

The correlogram represents the correlation between germination parameters and the growth parameters of the genotypes under study. Correlation among the 10 parameters at the germination stage with 5 parameters at the vegetative and 5 parameters at the reproductive stage was estimated by correlation analysis (Table 1) The germination percentage (GP) was found to be highly correlated to mean germination Time (MGT) (0.94) and negatively to mean germination rate (MGR) (−0.4). Fifty percent Emergence (50E) had the highest correlation to MGR (0.68) and was negatively correlated to Germination Index (GI) (−0.01). Seedling Length (SL) had its highest correlation to Seedling vigor (SV) and negative correlation to MGR. Plant Height (PH) showed the highest correlation to DFF (0.7) and the least to MGR (0.1). DFF showed the highest correlation to PH (0.7) and was not correlated to MGR. Pods per plant, seeds/pod, and 100 g weight showed a positive correlation with Yield, (0.71), (0.7), and (0.7) respectively, and the least or negative correlation with MGR (0), (−0.1), and (−0.2).

**Table 1.** Correlation of analysis of phenotypic parameters at vegetative, reproductive stages. Two-tailed correlation was determined from phenotypic data of germination, vegetative, and reproductive stages. PL.HT: Plant Height, TW: 100 Test weight, SV: Seedling Vigor, SL: Shoot Length, CVG: Coefficient of Velocity of Germination, MGT: Mean Germination Time, GRI: Germination Rate Index, GI: Germination Index, GP: Germination Percentage, MGR: Mean Germination Rate, 50%ET: 50% Emergence time. The color indicates correlation pattern, Red color: Negative correlation, Green color: Positive correlation.

| | GP | 50%ET | SL | SV | GI | GRI | MGT | CVG | MGR | TI | PL. HT. | DT 50% F | Pods/Plants | Seeds/Pods | TW | Y/Plant |
|---|---|---|---|---|---|---|---|---|---|---|---|---|---|---|---|---|
| **GP** | 1.0 | | | | | | | | | | | | | | | |
| **50%ET** | 0.1 | 1.0 | | | | | | | | | | | | | | |
| **SL** | 0.7 | 0.2 | 1.0 | | | | | | | | | | | | | |
| **SV** | 0.8 | 0.0 | 0.9 | 1.0 | | | | | | | | | | | | |
| **GI** | 0.8 | −0.1 | 0.5 | 0.6 | 1.0 | | | | | | | | | | | |
| **GRI** | 0.8 | −0.1 | 0.5 | 0.7 | 1.0 | 1.0 | | | | | | | | | | |
| **MGT** | 0.9 | 0.0 | 0.6 | 0.8 | 1.0 | 1.0 | 1.0 | | | | | | | | | |
| **CVG** | 0.9 | 0.0 | 0.6 | 0.8 | 1.0 | 1.0 | 1.0 | 1.0 | | | | | | | | |
| **MGR** | −0.4 | 0.7 | −0.2 | −0.4 | −0.5 | −0.5 | −0.5 | −0.5 | 1.0 | | | | | | | |
| **TI** | 0.9 | 0.0 | 0.6 | 0.7 | 1.0 | 1.0 | 1.0 | 1.0 | −0.5 | 1.0 | | | | | | |
| **PL. HT.** | 0.5 | 0.6 | 0.5 | 0.4 | 0.4 | 0.4 | 0.5 | 0.5 | 0.2 | 0.4 | 1.0 | | | | | |
| **DT 50% F** | 0.7 | 0.6 | 0.5 | 0.4 | 0.5 | 0.5 | 0.6 | 0.6 | 0.1 | 0.6 | 0.7 | 1.0 | | | | |
| **Pods/Plants** | 0.6 | 0.4 | 0.4 | 0.3 | 0.5 | 0.5 | 0.5 | 0.5 | 0.0 | 0.5 | 0.5 | 0.7 | 1.0 | | | |
| **Seeds/Pods** | 0.7 | 0.3 | 0.7 | 0.6 | 0.5 | 0.6 | 0.7 | 0.7 | −0.2 | 0.6 | 0.5 | 0.5 | 0.6 | 1.0 | | |
| **TW** | 0.4 | 0.5 | 0.4 | 0.3 | 0.3 | 0.3 | 0.4 | 0.4 | 0.1 | 0.3 | 0.6 | 0.6 | 0.5 | 0.4 | 1.0 | |
| **Y/Plant** | 0.6 | 0.3 | 0.5 | 0.4 | 0.4 | 0.4 | 0.5 | 0.5 | −0.2 | 0.5 | 0.5 | 0.4 | 0.7 | 0.7 | 0.7 | 1.0 |

*3.4. Principal Component Analysis (PCA)*

Analysis of the contribution of traits including 16 traits at both the CC and CS conditions among 204 genotypes throughout the plant life cycle was displayed in the screen plot with eigen values in *Y*-axis explaining the variation of each principal component captured (Figure 3). It contributed 96% and 97% of variability under CC and CS respectively, which could be explained by the first six components (PC1 to PC6). We observed a cutting-off point at PC2 and PC3 both in the control and cold stress conditions. After dimensionality reduction, while thePC1 and PC2 could describe only 75.2% of the data under CC (Supplementary Figure S1A), they could describe 76.4% of the data under CS (Supplementary Figure S1B). PC1 and PC2 explain 64.4% and 10.8% variance in data under CC (Figure 3A,B); and 57.7% and 18.7% of the variance in data under CS respectively (Figure 3C,D). These results indicated the reliability of the data for further analysis. PCA analysis showed data dimensionality reduction and highlighted the traits that contributed most to the total variance. While all the traits were found in the second and the third quadrant under CC, the traits were relocated in the first, third, and fourth quadrants under CS indicating a shift in their contribution under CS. The first quadrant (Q1) has the trait contribution in the order MGT > CVG > TI > GP > GRI > GI > SV. These results indicated that MGT could be the key contributing factor for cold resistance and the contributions of traits taper down to the least for SV. The third quadrant (Q3) has MGR. The fourth quadrant has its contributing variables in the order of 50E > PH > DFF > Seeds/pods > 100GW > Pods/plant > Yield/plant > SL. Thus, for the cold susceptibility, MGR was found to be the highest influencing trait followed by 50E and SL influencing the least. With the change of conditions from CC to CS during germination, the MGR in quadrant Q2 shifted to Q3. While the Yield/plant, PH, SL, DFF. 50E, Pods/plant, Seeds/pods, and 100GW shifted from Q2 to Q4. GP and SV shifted from Q2 to Q1. MGT, CVG, TI, GI, and GRI shifted from Q3 to Q1. These results indicated that GP, SV, MGT, CVG, TI ad GRI are the key contributing traits for cold resistance. The genotypes G6, G12, G33, G34, G35, G37, G40, G44, G52, G53, G69, G74, G77, G88, G135, G153, G154, G155, G158, G164, G169, G195, G197 (Accession codes of these genotypes are present in Supplementary Table S1)fell in the positive quadrants of PC1 and PC2, could be considered as cold tolerant genotypes. On the other hand, the genotypes, G8, G10, G20, G28, G57, G71, G73, G84, G85, G92, G93, G99, G111, G113, G126, G128, G129, G130, G143, G145, G160, G193 (Accession codes of these genotypes are present in Supplementary Table S1) falling in the negative quadrants of the PC1 and PC2 could be considered as the susceptible genotypes. The rest of the genotypes were categorized as intermediates in terms of cold resistance.

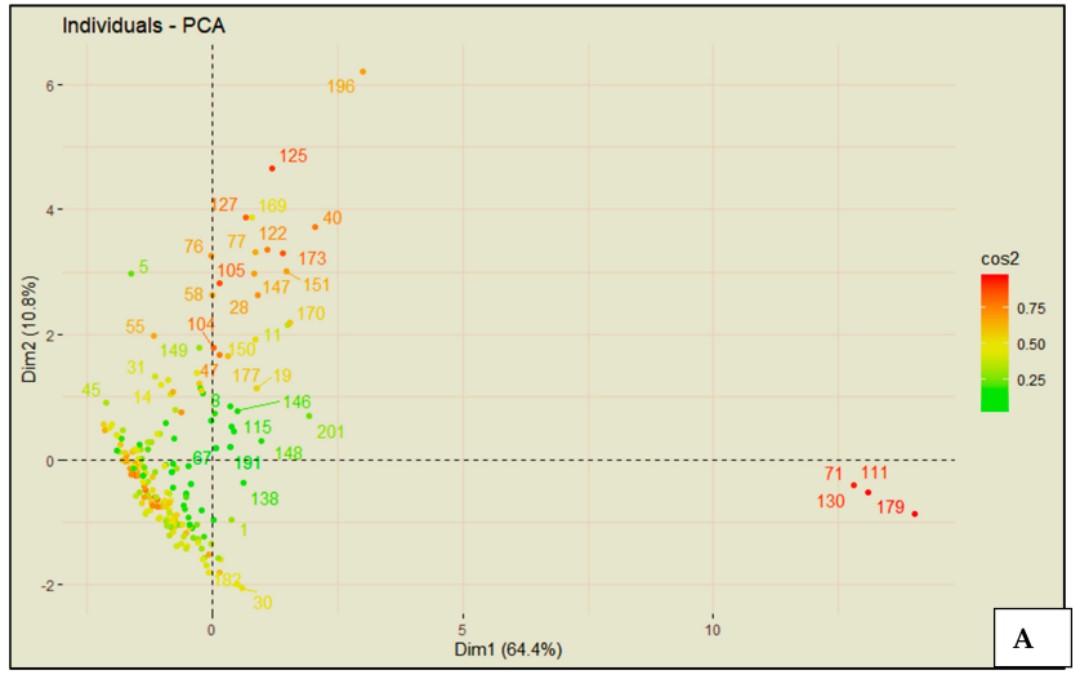

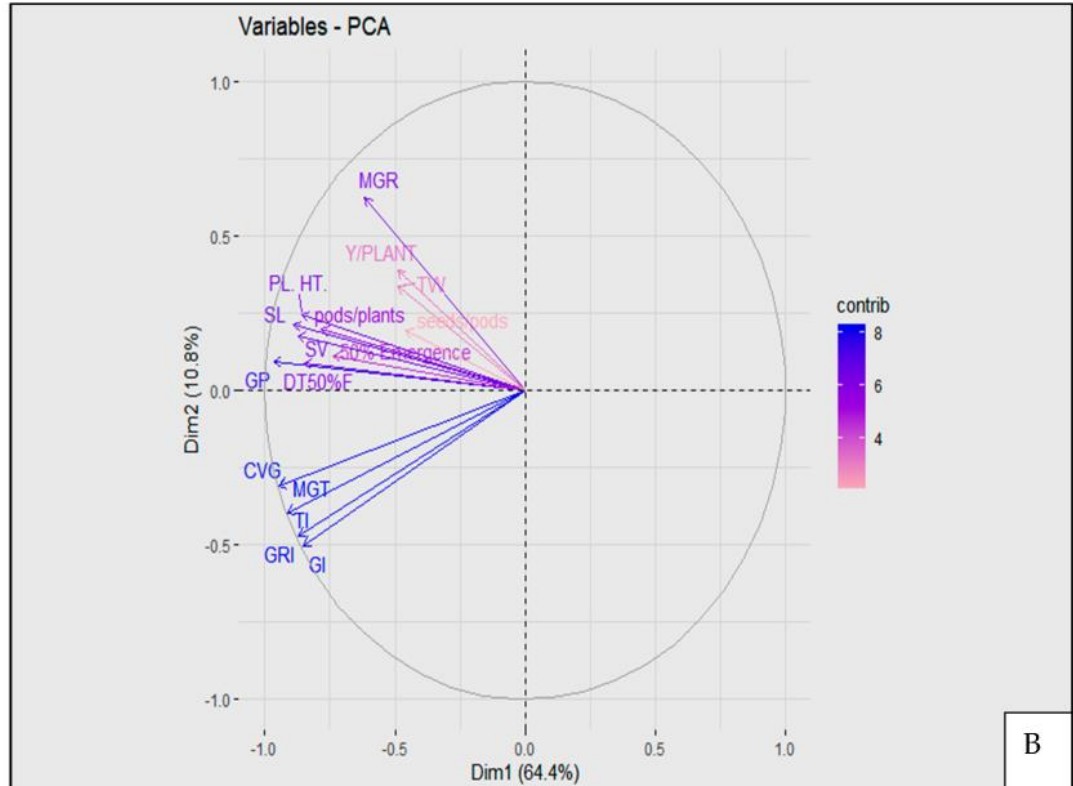

**Figure 3.** *Cont.*

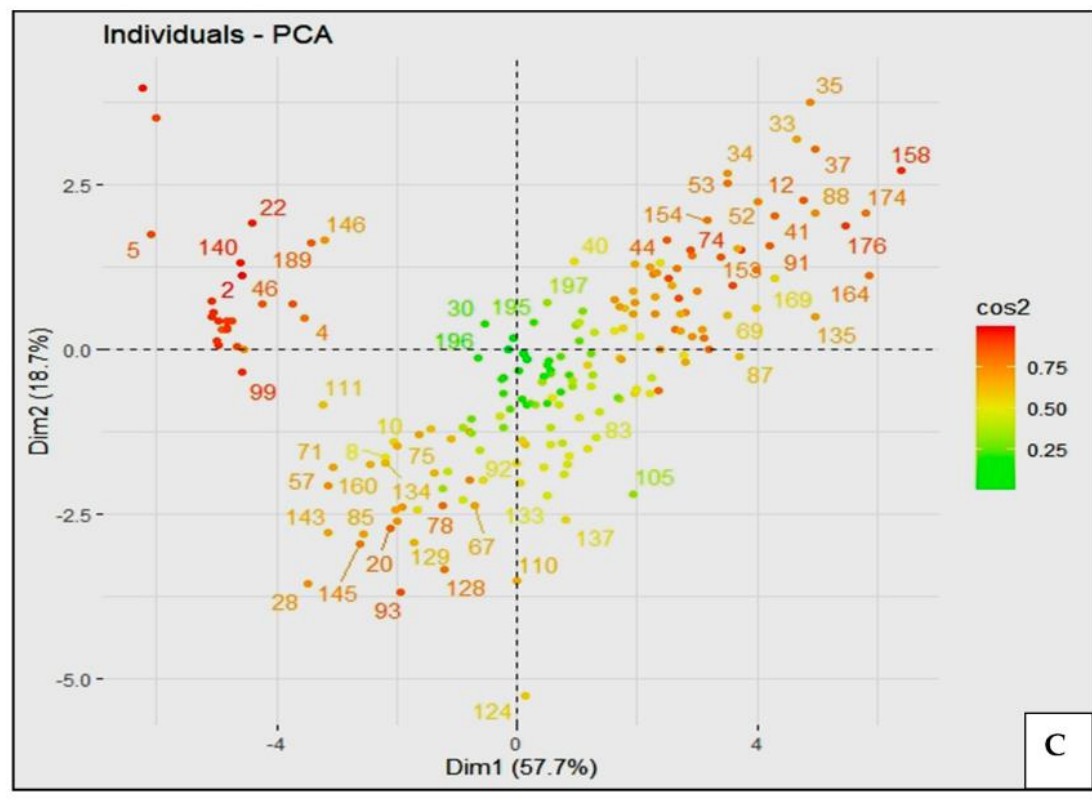

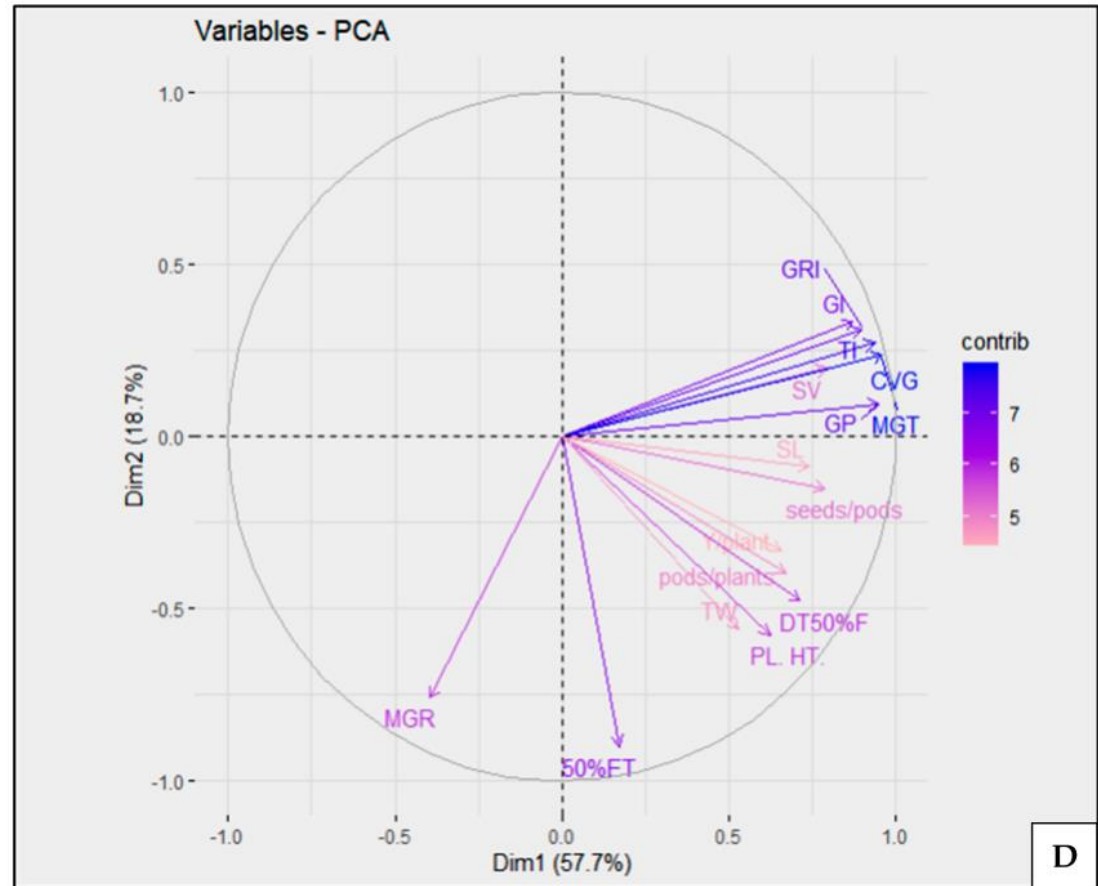

**Figure 3.** Individual PCA plots showing clusters based on similarity of samples. PCA analysis at
(**A**,**B**) 22 °C and (**C**,**D**) 10 °C. The quality of representation of the variables on factor map is indicated
by cos2 (square cosine, squared coordinates). A cos2 value correlates with the color and resistance to

cold accordingly if present in positive quadrant and vice-versa. Dim indicates Dimensions/Principal components. The red dots represent the genotypes and the grey arrow represents the traits or the variables that are taken into consideration.

### 3.5. Clustering Analysis of 204 V. radiata Genotypes Based on Derived Stress Indices

To reclaim the accuracy of the results obtained from the PCA analysis regarding the grouping of the genotypes, K-means clustering based on five derived stress indices (i.e., GP, GSTI, SHSI, RCI, and PCI) was applied to generate a dendrogram and the 204 genotypes when cold treatment was given at the seedling stage and could be grouped into 3 clades (Figure 4). The dendrogram generated had three clades. The analyses of these stress indices among the genotypes in the three clades are divergently represented in the violin plots. Clade I consist of 135 genotypes, which contained a group of genotypes with most of 5 parameters in the lower range, indicating them to be cold susceptible. On the other hand, Clade II comprises 58 genotypes, and the values of 5 stress parameters are not consistent in the intermediate range in all cases. Clade III comprises 11 genotypes with most of 5 parameters in the higher range, indicating them to be cold resistant. These lines were cold tolerant according to GP, GSTI, and SHSI, but RCI and PCI were in the intermediate range and, hence, were grouped as immediately cold-tolerant genotypes. These results indicated that the derived stress parameters are highest in the resistant group of lines compared to the susceptible group of lines. The resistant clade III had three clusters. The susceptible clade I comprises two divergent sub-clades ultimately subdivided into nine clusters. Whereas the intermediate clade II was apparent as 2 closely related sub-clades, one of which is a small cluster consisting of ~15 genotypes, excluding the bigger cluster with ~43 genotypes. This result indicated that the response of the genotypes to cold stress has evolved differently among the susceptible, intermediate, and resistant clusters. The grouping of the 204 genotypes for cold resistance according to the K-means clustering-based dendrogram matched the grouping of genotypes according to PCA analysis. These results confirmed the reliability of the methods used for analysis in this study and for further characterization of the resistant, intermediate, and susceptible genotypes.

### 3.6. Seedling Phenotype of Selected Lines for Response to Cold Stress

Based on the positioning of the genotype in the bioplot (their occurrence in the quadrants under CC and CS) and grouping of genotypes in the dendrogram, G77 (PAU 911), G92 (PUSA 9531) could be selected as the representatives for resistant and susceptible genotypes respectively in response to cold stress. G88 (PUSA 1672) had inconsistent positioning in the biplot and a similar trend in the case of stress analysis, and hence it was taken as representative of the intermediate cold response genotype. These three lines showed 81 to 90% of germination respectively under CS (indicated as colored arrows in Figure 1D. These lines were analyzed for their seedling phenotype and biochemical and molecular characteristics under CS.

Root length (Figure 5A) and seedling length (Figure 5C) showed significant ($p < 0.0001$) reduction in response to eight-day cold in all the genotypes, whereas, in the case of rootlet number (RLN) (Figure 5B) and leaf area (LA) (Figure 5D), the highest reduction was observed in the G92 (found to be susceptible previously), and the least reduction in G77 is considered as resistant in previous categorizations. G88 had intermediate values in the case of RLN and LA, which matched its previous categorization under intermediately cold resistance.

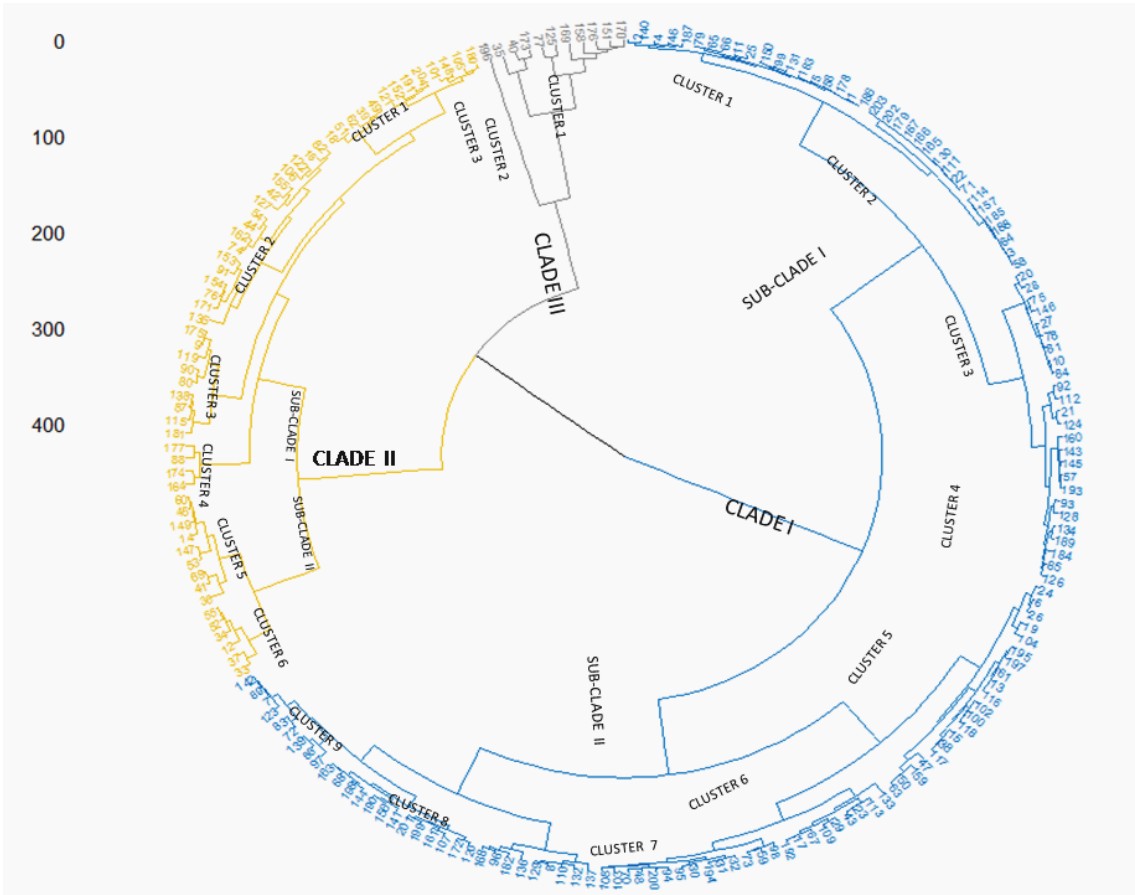

**Figure 4.** Hierarchical clustering of 204 *V. radiata* genotypes based on derived stress indices. Four derived stress analysis parameters (GP: Germination percentage, Germination stress tolerance index: GSTI, SHSI: Seedling height shoot index, PCI: Productive capacity index, RCI: Resilience capacity index; formulas for calculation of these indices are present in Materials & Methods) during germination, reproductive and post-harvest phase of growth were computed in 204 *Vigna radiata* genotypes at 22 °C or 10 °C. These genotypes were further grouped into three subclusters in a dendrogram according to degree of resistance to 10 °C using "ward" method [44]. Highest agglomerative coefficient observed = 0.9913957. The 1st subcluster: 135 genotypes (Blue color), the 2nd subcluster: 58 genotypes (Yellow color), and the 3rd subcluster: 11 genotypes (Grey color).

### 3.7. Physiological and Biochemical Characteristics of Selected Lines with Response to Cold

The stress of eight days of cold could lead to a significant reduction in the total chlorophyll (Chl) as well as the Chl a, b pigments in all the 3 genotypes, with the highest in the G88 (PUSA 1672) under CS intermediate genotype (Supplementary Figure S2A). Similarly, carotenoid content (Supplementary Figure S2B) was also the least in G92 (PUSA 9531) under CC and highest in G88 (PUSA 1672) after CS. Total Chl content (Supplementary Figure S2A) was found to be least in the G92 (PUSA 9531) in CC, with a reverse trend of higher Chl A than Chl B to that of the G77 (PAU 911) and G88 (PUSA 1672). This result indicated the CS susceptible genotype had less chlorophyll and pale green leaves than the CS-resistant and intermediate genotypes under normal growth CC. This indicated that the least effect of CS on Chl and carotenoid was observed in the CS intermediate G88. Relative water content (Supplementary Figure S2C) was affected only in G88 (PUSA 1672) after CS, with nearly no effect in G77 (PAU 911) and G92 (PUSA 9531). Malondialdehyde (MDA) (Supplementary Figure S2D) and sugar content (Supplementary Figure S2E) tend to increase in all the 3 genotypes with an effect onCS. The highest increase in MDA and sugar content was observed in G88 after CS. Reactive oxygen species and their detoxification during CS were assayed with the help of nitro blue tetrazolium (NBT) staining to stain the

$O^{2-}$ radicals and superoxide dismutase (SOD) enzyme activity assay, respectively. SOD enzyme actively increased with effect to CS in the 3 genotypes with the highest in G92 (PUSA 9531), which was the CS susceptible genotype (Supplementary Figure S2F). The $O^{2-}$ radicals were found to be decreased under CS, and at least $O^{2-}$ was also found in G92 (PUSA 9531) (Supplementary Figure S2G). The highest increase in proline content was observed in the resistant genotype (G77) after stress (Supplementary Figure S2H). Similarly, the percentage of electrolyte leakage was also the least in G77 (PAU 911) under CC and highest in G92 (PUSA 9531) after CS (Supplementary Figure S2I). The index of injury was maximum in the susceptible one (G92) compared with the registrant (G77) and intermediate genotype (G88) (Supplementary Figure S2J).

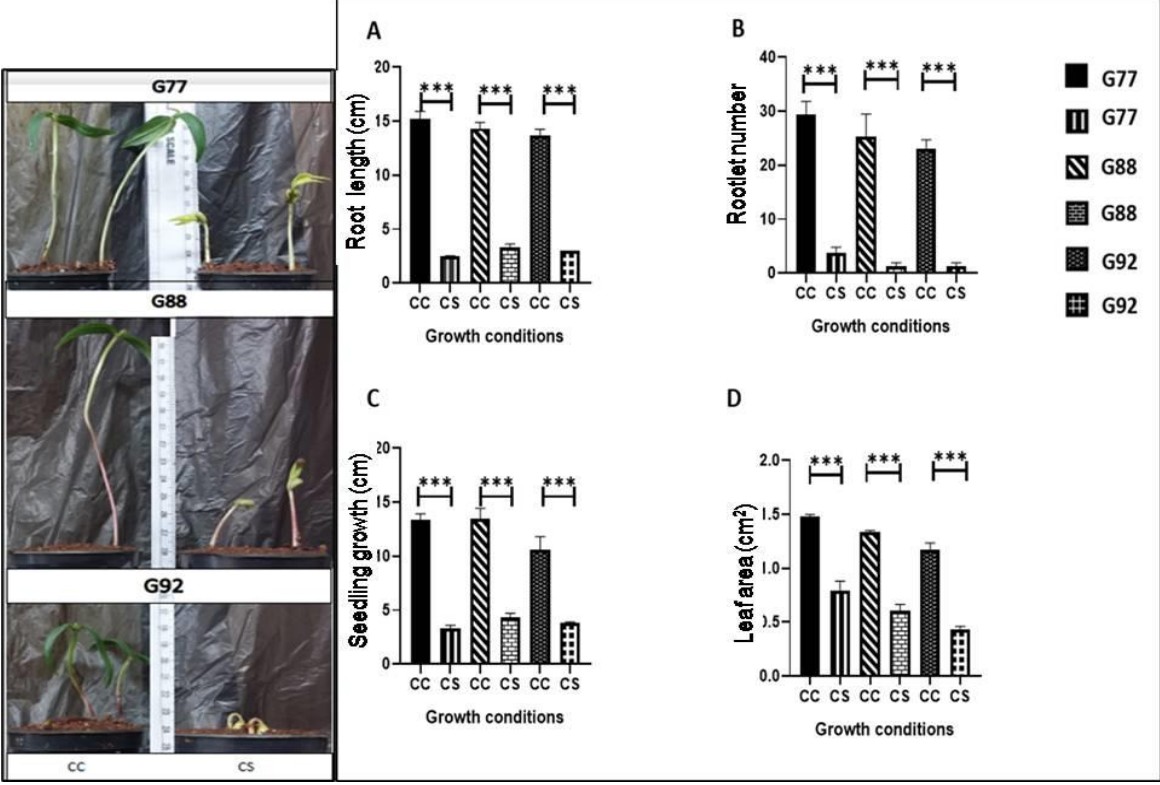

**Figure 5.** Seedling phenotypes of G-77 (PAU 911), G-88 (PUSA 1672), and G-92 (PUSA 9531) under Cold Stress (CS). Seedling phenotypes (picture panel) such as (**A**)Root length (**B**) Rootlet number and (**C**) Seedling length (**D**) Leaf area were measured in 8-day-old seedlings grown at 10 °C. Data represented were mean of 10 measurements for each with 3 biological replicates. Statistical significance using one-way ANOVA indicated as *** $p < 0.0001$.

### 3.8. Anatomical Analysis

On the basis of the results of confocal microscopic analyses, it was indicated that the stomatal density per unit area increased under CS as compared to CC, with the exception of no change in G92 (PUSA 9531) (Figure 6A). The highest (45.34) stomatal density under CS was observed in the intermediate genotype G88 (PUSA 1672) followed by resistant G77 (PAU 911) (35.12) and susceptible G92 (PUSA9531) (28.53) (Figure 6A). Contrastingly, the open pore % decreased with effect to CS, again with the exception of no change in G92 (PUSA 9531) (Figure 6B). During CS, the open pore % was the highest in the susceptible genotype G92 (PUSA 9531) (75.23%), followed by intermediate G88 (PUSA1672) (25.19%) and resistant genotype G77 (PAU911) (20.48%) (Figure 6B). The trichome density increased under CS in all the 3 genotypes with the highest being present in the susceptible genotype G92 (PUSA 9531) (10.23) followed by resistant G77 (PAU911) (9.3) and intermediate genotype G88 (PUSA1672) (7.11) (Figure 6C). Autofluorescence during confocal microscopy

indicates the presence of secondary metabolites in the sample of analysis [43]. There was a significant increase in the autofluorescence intensity under CS in all three genotypes. The autofluorescence intensity% was found to be highest intensity in resistant G77 (PAU911) (90%) and was at par with an intermediate G88 (PUSA1672) (89%), followed by susceptible genotype G92 (PUSA 9531) (70%) (Figure 6D). There was an increase in the mesophyll cells per unit area in all three genotypes under CS with an appearance of thickly packed cells and nearly invisible air spaces. The greatest number of cells per unit area was observed in the intermediate genotype G88 (PUSA 1672) (212.15), followed by the susceptible G92 (PUSA9531) (170.19) and was the least in the resistant one G77 (PAU911) (149.12) (Figure 6E).

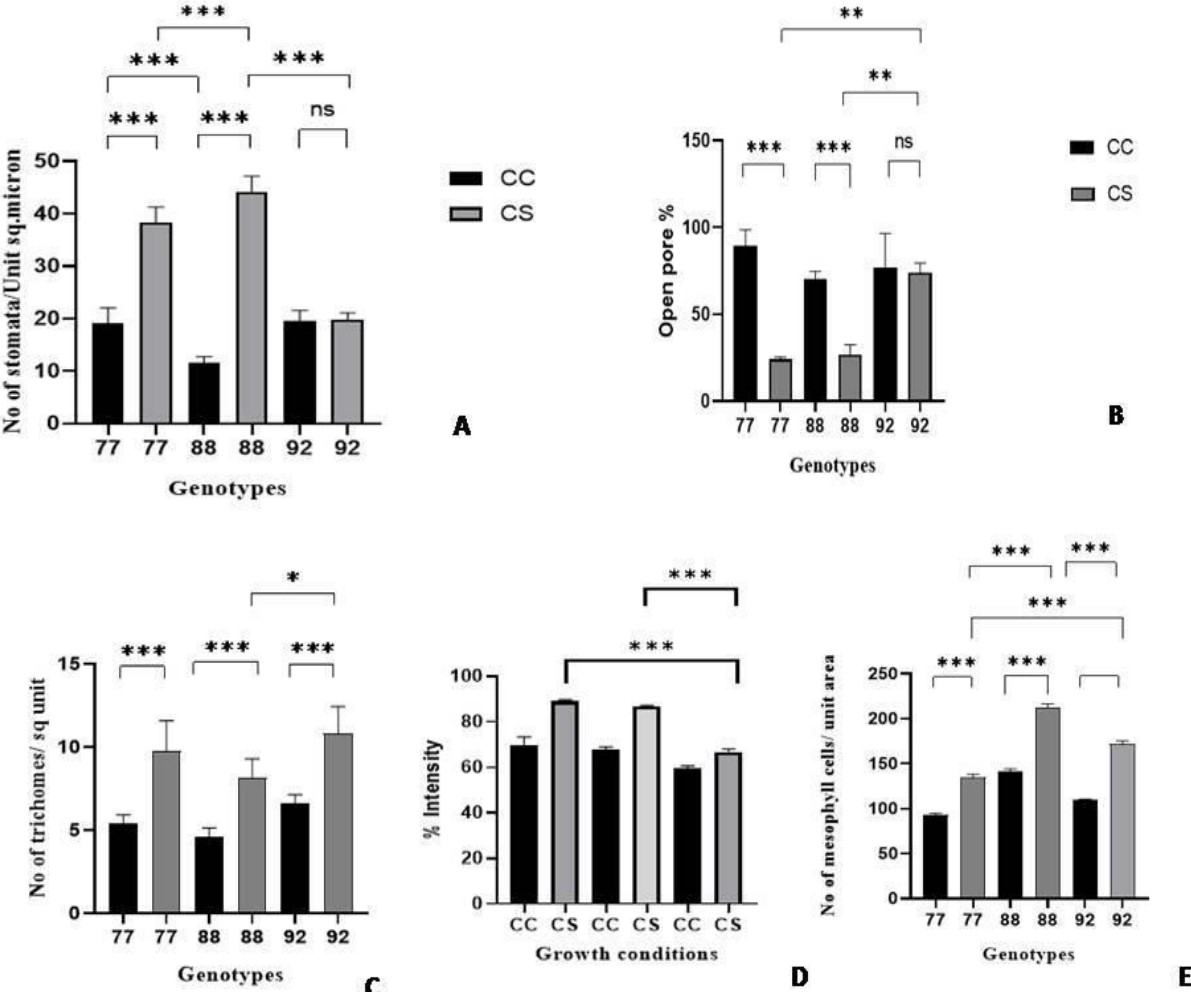

**Figure 6.** Microscopic analysis of stomata and trichomes in response to cold stress in selected *V. radiata* genotypes. stomatal density (**A**), stomata open pore percentage (**B**), trichome density (**C**), Autofluorescence analysis (**D**) Number of mesophyll cells per unit area (**E**) in the selected genotypes (G77) PAU911, (G88) PUSA1672 and (G92) PUSA 9531) of *Vigna radiata* grown under cold stress at 10 °C (CS) and control condition 22 °C (CC). All the data are analyzed on the basis of one-way ANOVA. * $p < 0.01$, ** $p < 0.001$, *** $p < 0.0001$.

### 3.9. Protein Estimation and MS Analysis on Cold Stress

Total protein extracts from the 3 genotypes were separated in a denaturing PAGE and showed different banding patterns after CS conditions (Figure 7A). While the total amount of protein reduced non-significantly after CS in the resistant genotype G77 (PAU911), it was with negligible change in G88 (PUSA1672) and increased insignificantly in G92 (PUSA 9531) (Figure 7B). The total amount of protein left after CS was the highest in G77 followed

by that in the intermediate (G88), and was the least in the susceptible (G92) genotype. The protein band from G77 (PAU911) at ~75 kDa under CC appeared to have a shift to higher molecular mass under CS (B1, black arrow). Whereas a specific band of G88 (PUSA1672) ~70 kDa which was found under CS condition was absent under CC (B2, Red Arrow). The MS analysis of these 2 bands revealed that there was no relevant similarity of B2 with abiotic stress response, whereas B1 showed to have a match with *TETRATRICOPEPTIDE-REPEAT THIOREDOXIN-LIKE1* (*TTL1*) (UNIPROT Identifier: LOC106762419) with 41% sequence coverage, E-value 0.023 (Figure 8).

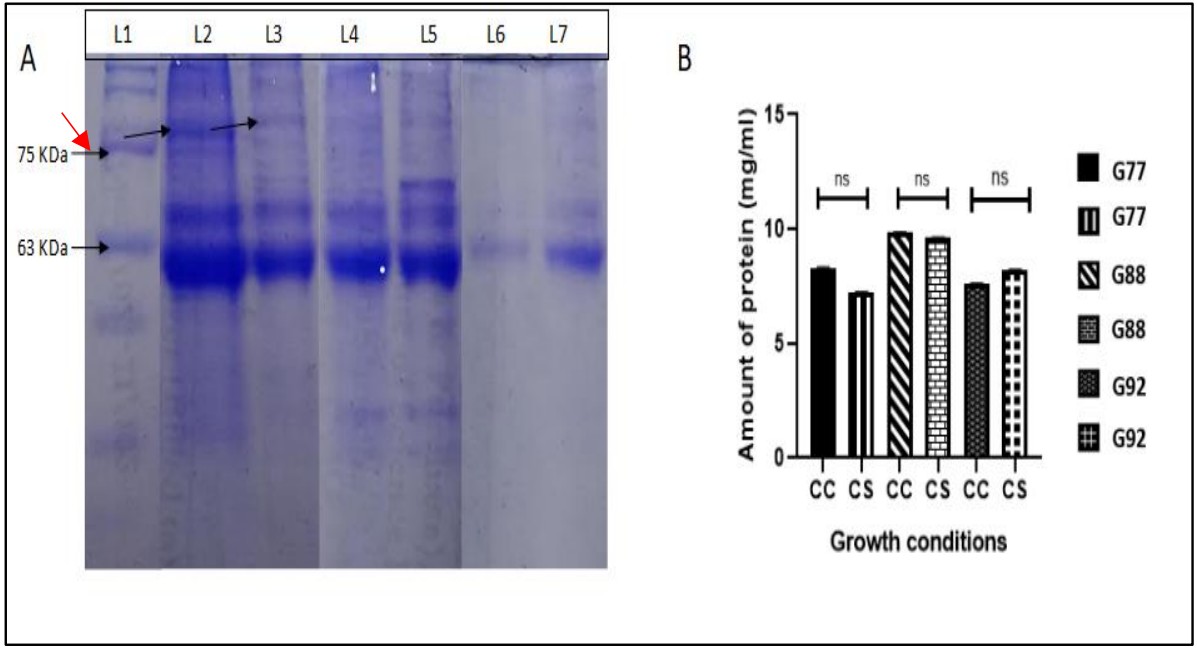

**Figure 7.** Total protein estimation in the selected genotypes after CS. (**A**) Total protein extracts were separated in a SDS-PAGE. L1-Protein marker L2-G77 at CC, L3-G77 at CS, L4-G88 at CC, L5-G88 at CS, L6-G92 at CC, L7-G92 at CS. Black arrow in L2 and L3 indicate B1 (~75kDa) from G77. Red arrow in L5 indicate B2 (~70 kDa) from G88 (**B**) total protein quantification was carried out using amido black method from 8-day-old seedlings grown at CS. G77, G88 and G92 are the genotypes, PAU911, PUSA1672 and PUSA9531 respectively. Statistical significance using one-way ANOVA indicated as ns: Non-significant.

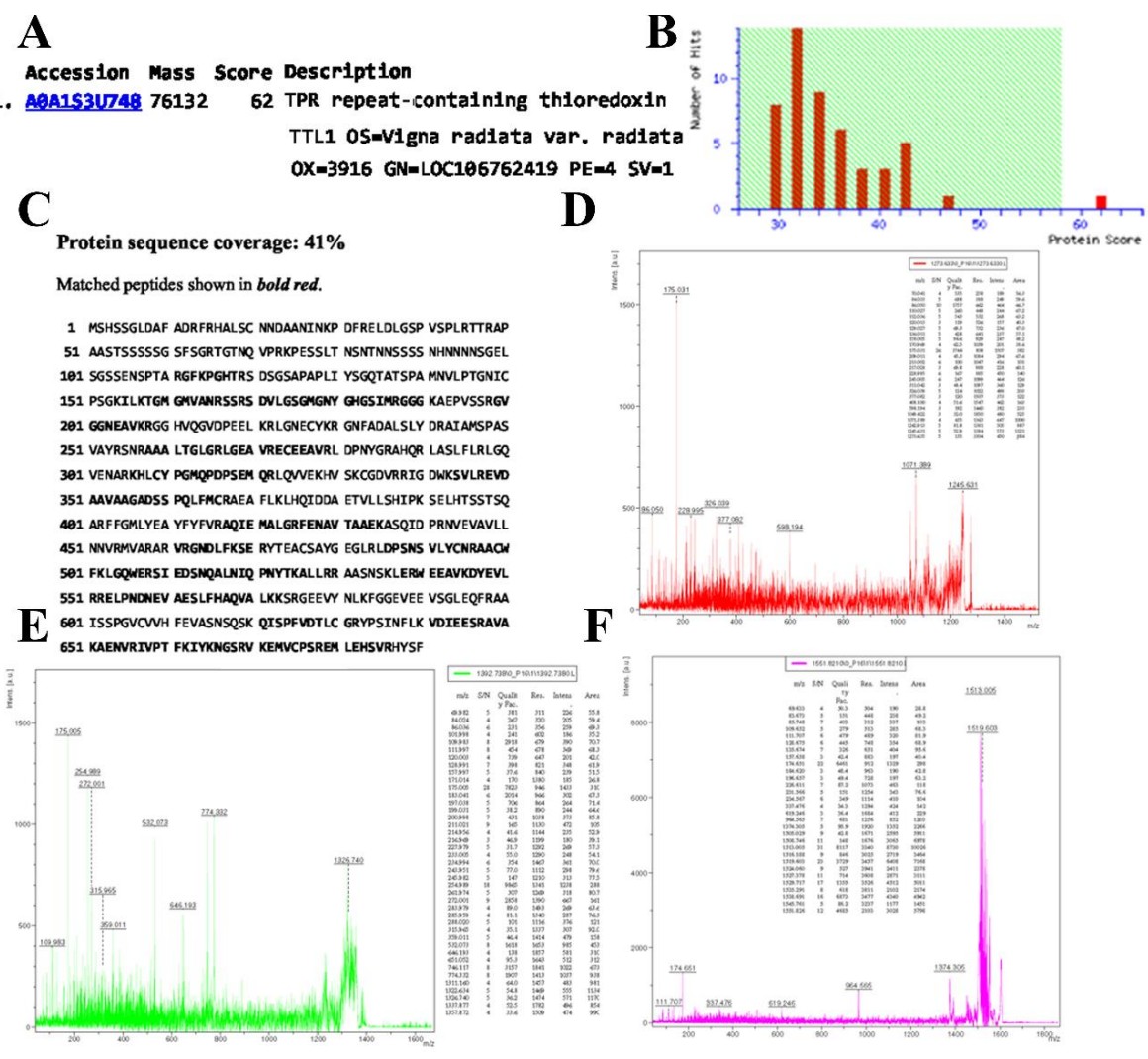

**Figure 8.** MS analysis result of cold-resistant band from PAU911 (G77). Protein-containing gel bands obtained after SDS-PAGE were analyzed using MALDI TOF/TOF MS. The peptide stretches were obtained after trypsin digestion at C-terminal side of KR unless the next residue is P. Good quality peptide sequences were searched on MASCOT database and for hits using BLASTX in NCBI database using parameters described in Section 2.7. (**A**) MASCOT search result for B1 from G77 in database: *Vigna radiata*_Mungbean (35,422 sequences; 16,509,319 residues) with Top score:62, Accession: A0A1S3U748. (**B**) MASCOT score histogram: Protein score is $-10 \times \log(P)$, where P is the probability that the observed match is a random event. Protein scores <58 are significant ($p < 0.05$). (**C**) MASCOT search result for A0A1S3U748: *TETRATRICOPEPTIDE-REPEAT THIOREDOXIN-LIKE1* (*TTL1*)OS = Vigna radiata var. radiata OX = 3916, GN = LOC106762419 PE = 4 SV = 1; Expect = 0.023, Nominal Mass:76,132, pI:8.87. Sequence similarity is available at NCBI BLAST to search for A0A1S3U748 against nr. Protein sequence coverage:41%, Matched peptides are shown in Bold and Yellow highlights **Bold** (**D**–**F**) Protein sample spectra at 1273aa, 1392aa, and 1551aa. The *m/z*, S/N, quality of the fraction, residue, intensity and area of the peaks are shown in column view inside each graph.

## 4. Discussion

Cold stress severely affects germination by delaying germination, poor emergence, seedling establishment, reduced plant density, and uneven stand establishment. Our study presented an approach based on germination parameters to select cold-tolerant mungbean genotypes. It also established a method for screening mungbean genotypes for cold stress response at the seedling stage, which indicates cold tolerance at vegetative

and reproductive stages with sustainable yield. Similar germination-based screening in rice showed promising results previously [45]. Our result showed that germination percentage along with the percentage of recovery rate could be proved as a strong, reliable parameter for early screening of cold tolerance during the germination stages, apart from the SV, GI, MGT, and TI. Surprisingly, cold stress of the initial 8-days during seedling establishment imposed a negative impact throughout the life of mung bean genotypes, which were reflected in different parameters of the vegetative stage such as plant growth including plant height, DFF, and pods/plant as well as in the reproductive stage such as seeds/pod, yield/plant, and 100-seed weight. Correlation analysis showed a distinct correlation between the seedling traits and mature plant traits separately. However, PCA and biplot analysis could bring better resolution in the selection of traits and grouping of genotypes. The vectors of the biplot indicated GRI, TI, MGT, and CGV to be more influential than other germination parameters. In addition, 50E and DFF were found to contribute higher than the other parameters in this study. Our phenotyping revealed that the resistant and intermediate genotypes had a morphological and phenotypic advantage over the susceptible ones under cold stress. Along with this clarity on traits for cold response screening, our study selected 11 cold tolerant and 135 cold susceptible *V. radiata* genotypes, which could be used as a reliable source for understanding cold stress tolerance at the protein level. The cold-tolerant *V. radiata* would be used for further testing under field study. Derived stress parameter-based analysis could facilitate effectively grouping the whole 204 genotypes into 3 clusters, as a grouping of almost 85% of the genotypes coincided with the results of the biplot analysis. Among the physiological and biochemical characteristics studied, chlorophyll, carotenoid, and MDA content may appropriately represent the cold response. Cold-induced photosynthetic inhibition is due to reduced chlorophyll synthesis, poor chloroplast development, diminished efficiency of photosynthetic apparatus, restricted carbohydrate transport, limited stomatal conductivity, suppressed Rubisco activity during carbon assimilation, disrupted electron transport chain, proline content and decreased energy stock [46,47]. These were evident with a better survival method as observed in the resistant genotype G77 (PAU 911), susceptible G92 (PUSA (9531) and intermediate G88 (PUSA 1672). On the basis of intraclustral analysis of derived stress parameters, it was revealed that each of the clusters had more pronounced effect of stress during the germination rather than the reproductive and vegetative stage. It also inferred that the cold stress during germination could have a considerable influence on the vegetative and reproductive fitness of the plant. Seedling phenotype reclaimed that the resistant and intermediate genotype has better stress tolerance in terms of rootlet umber than the susceptible one. The physiological and biochemical analysis also revealed similar cold tolerance responses over the susceptible ones during cold stress. Confocal microscopic analysis revealed that the resistant genotype (PAU 911) has the least open stomatal pore percentage as compared to intermediate and susceptible ones. The trichome density was higher in susceptible ones under stress conditions. When a plant undergoes cold stress, several morphological alterations occur [48] that include root-shoot growth being hampered with reduced productivity. Prolonged exposure to cold stress results in stunted growth, diminished root-shoot surface area, leaf chlorosis, and disturbed water and nutrient relations. Such indicators lead to a significant reduction in yield and quality [49]. There are several different strategies to show CS response in the 3 genotypes. G77 (PAU 911) though had the highest amount of protein after CS, its reproductive parameters were compensated. Whereas G88 (PUSA 1672) was categorized in intermediate genotype possibly for it could retain a higher amount of protein left after CS, G88 (PUSA 1672) had equivalent SOD activity as compared to the G77 (PAU 911). Results of the present study could able to identify a *TETRATRICOPEPTIDE-REPEAT THIOREDOXIN-LIKE1* (*TTL1*) for cold tolerance specifically in the resistant G77 (PAU 911). It was found to be of 690 amino acids and 76.13 kDa molecular mass. *Arabidopsis thaliana* TTL1 has been found to be associated with osmotic stress tolerance and linked to the abscisic acid pathway [50]. The TTL1 is already been shown to be involved in osmotic stress tolerance in *A. thaliana*. However, it could

also have implications for cold stress response in resistant genotypes of *V. radiata* as shown in our study. Cold stress induces a complex network of signaling pathways mediated by transcription factors, plant hormones, reactive oxygen species (ROS), and other primary and secondary messengers. ROS are generated as a result of aerobic metabolism in plants. Due to their unstable nature, these ROS can damage various cellular components. In order to avoid such damage, plants have developed various redox regulatory systems. Interestingly, ROS are purposefully produced to serve as messengers by plants when they are under cold stress. ROS are long known as cytotoxic molecules that damage cellular metabolism [5]. ROS have dual functions first with a loop dependent on the dosage, at a low level acting to trigger defenses and developmental responses at early stages, and second with a high level attacking the cell membrane for destroying the cell. ROS regulates the trade-off between growth and defense, where a central transcriptional network regulates the growth and defense pathways to balance the fate of the plant. Therefore, stresses can trigger oxidative stress, and thus, limit vegetative growth and reproduction capacity [51]. On the basis of MDA content during different stages of plant development, it is evident that the adverse effects are accompanied by structural alterations in the membrane [52], which were subsequently followed by cellular leakage of electrolytes and amino acids, diversion of electron flow toward alternate pathways [53], alterations in protoplasmic streaming, and re-distribution of intracellular calcium ions. These symptoms are directly correlated with injury to membrane structures of cells and changed lipid composition. Cold-induced alterations in crop plants in particular lead to decreased ATP synthase activity, followed by inhibition of Rubisco regeneration and photophosphorylation [54]. Cold-induced photo-inhibition subsequently leads to a reduction in photosynthetic activity [55,56]. If cold stress remained for a shorter duration, plants could recover their normal state, but such a situation is irreversible under a prolonged cold duration. The prolonged cold stress period causes severe damage to the mitochondrial structure, slows down the flow of kinetic energy, and disrupts enzymatic activity, ultimately diminishing the respiration rate [57,58].

## 5. Summary

The present study based on germination parameters establishes a method for screening mungbean genotypes (204 genotypes in the present study) for cold stress response at the seedling stage, which can indicate cold tolerance at vegetative and reproductive stages with sustainable yield. The screening was conducted using germination parameters such as % germination, 50% emergence time, seedling height, seedling length, seedling vigor, germination rate index, mean germination time, coefficient of the velocity of germination, mean germination rate, and Timson's germination index computed from the data obtained till 8th day after sowing. Categorization of germplasms into resistant, intermediate, and susceptible cold response was conducted using derived stress indices such as Promptness Index, Germination stress index, Plant Height stress index, Stress Susceptibility Index, Mean Productivity Index, Stress Tolerance Index, Geometric Mean Productivity, Resilience Capacity index and Productive Capacity Index to the life cycle. Three genotypes (i.e., PAU911 (G77), PUSA 1672 (G88), and PUSA 9531 (G92)) were selected for showing resistant, intermediate and susceptible cold response respectively on the basis of the above-mentioned parameters and further characterization of their phenotypic, physiological, biochemical and molecular characteristics. On the basis of phenotyping, it is revealed that the resistant and intermediate genotypes had higher advantages than the susceptible ones under cold stress. The resistant PAU911 (G77) genotype was distinguished for having a higher rootlet number, leaf area, and increased chlorophyll, carotenoid, and MDA content at 10 °C. Moreover, it had the least open stomatal pore percentage, reduced trichome density, and higher secondary metabolite content compared to the susceptible genotype (PUSA 9531 (G92)) under cold stress. This study identified *TETRATRICOPEPTIDE-REPEAT THIOREDOXIN-LIKE1* (*TTL1*) protein in the resistant (PAU 911) genotype under cold stress. This cold-indicative protein TTL1 has been shown to be involved in osmotic stress tolerance in *A. thaliana*. However, it may participate in signaling cross-talk as seen in cold-tolerant mungbean expressing this

protein. Understanding the mechanism of cold stress tolerance and signaling network is important for crop improvement.

**Supplementary Materials:** The following supporting information can be downloaded at: https://www.mdpi.com/article/10.3390/agriculture13020315/s1, Table S1: List of 204 germplasms used in the experiment; Table S2: The Formulas used for computing germination and stress parameters. NS: control condition, S: Cold condition, Y: yield, X; Figure S1: Scree plot displaying the variation of each principal component captured from the data taken at control conditions (A) and Cold stress (B). Dim indicates Dimensions/Principal components. PC1/Dim1 explains 64.4% and PC2/Dim2 explains 10.8% of the variance in data; Figure S2:Biochemical analysis of cold stress response in selected *V. radiata* genotypes (A) Chlorophyll content, (B) Carotenoid content, (C) Relative water content, (D) Lipid peroxidation-Malondialdehyde (MDA), (E) Total sugar content, (F) SOD enzyme activity (G) $O^{2-}$ Superoxide content, (H)Proline content, (I) Percentage of electrolyte leakage and (J) Index of injury were assayed in seedlings after germination and growth at 10 °C for 8 days. Data represented were mean of 10 measurements for each with 3 biological replicates. Statistical significance using one-way ANOVA indicated as *** $p < 0.0001$, ** $p < 0.001$ and * $p < 0.01$.

**Author Contributions:** L.S.M., G.R.R., K.C.P. and M.P. conceived and designed the experiments. L.S.M. conducted the experiments. G.M. and L.S.M. performed all microscopy experiments. G.R.R. provided the seed source and field experiment. K.C.P. provided an infrastructural facility for work. G.R.R., K.C.P. and S.K.P. provided funding support. L.S.M. and M.P. analyzed the data, L.S.M. prepared the figure and both co-wrote the paper. G.R.R. and S.K.P. edited the MS. All authors have read and agreed to the published version of the manuscript.

**Funding:** The APC was funded by S.K.Panda.

**Institutional Review Board Statement:** Not applicable. This study is not involving humans or animals.

**Data Availability Statement:** L.S.M & G.R.R will provide the data on a request basis.

**Acknowledgments:** The authors wish to acknowledge to the Pulses Center, Odisha University of Agriculture & Technology, Bhubaneswar & Department of Plant Pathology, Banaras Hindu University, Varanasi for proving the seeds for conducting the experiment. L.S.M is acknowledging to Indian Council of Agricultural Research for providing SRFfellowshipand also to School of Biological Science, NISER, Bhubaneswar for providing laboratory facility.

**Conflicts of Interest:** The authors declare no conflict of interest.

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
