# Peer review of "Cold Tolerance Mechanisms in Mungbean (Vigna radiata L.) Genotypes during Germination"

_agriculture, doi:10.3390/agriculture13020315_

Round 1

Reviewer 1 Report

This manuscript described evaluation for cold tolerance in 204 mungbean germplasms, screening high tolerant accessions and identification of a novel protein associated with cold tolerance in mungbean. The authors indicated that cold stress treatment induce negative effects to plant height, flowering time, the number of pods and sees, seed weight and final yield. Principal component analysis and phylogenic analysis revealed that germplasm accessions were divided into three major groups depending on difference levels of cold tolerance. Based on these results, the authors selected three mungbean accessions showing different cold tolerance levels. Additional experimental results indicated that strong cold tolerance accession showed higher rootlet number, leaf area, chlorophyll content, carotenoid content and MDA content. And, the authors identified one protein for TPR repeat containing thioredoxin TTL1, which was highly correlated with cold stress response in the mungbean accessions. The authors carried out a lot of experiments to obtain their conclusion in this study. However, the authors should make several revisions in the current version of this manuscript described below.

1) The authors should delete unnecessary spaces at the Title, Abstract and other section in the text.

2) L. 17, and L. 576-579. The authors evaluated cold stress as germination percentages at 10 oC at 8th day after sowing in this study. How did you develop this evaluation method? If you already reported development of this method in previous studies, you should indicate the citation for the previous studies.

3) L. 110. The authors should move Table 1 to other section in the text. Present position is not appropriate to indicate this result. I think that L. 382 would be a good position for showing these correlation results.

4) L. 102-109, L. 126-146, L. 153-158, Table 1, Table S2, L. 373-382, and other sections in the text. Abbreviations of trait names are inconsistent among several sentences. Some abbreviations were lost (germination percentage at L. 127 etc.), and some abbreviations were duplicated (SL indicated twice at L. 130 and L. 144 etc.). The current descriptions are very confusing. So, the authors should correct all of descriptions for abbreviations in the text.

5) L. 193. Please check the font for 'Panigrahy et al. 2018'.

6) L. 251. You should insert line break here.

7) L. 271-279. The composition of the protein extraction buffer should be described in writing as well as other solution in the text.

8) L. 311-333. The authors should indicate citations for each part in the text such as Fig. 1A, 1B, 1C, 1D and 1E. The same for the other figure citations of the description.

9) L. 334. Fig. 1 doesn't have descriptions for B and C.

10) L. 334. Can you indicate positions of three selected accessions G77-PAU911, G88-PUSA 1672 and G92-PUSA 9531 as different color arrows in Fig. 1D? And, the authors could also indicate the position of IPM-02-14 in Fig. 1E.

11) L. 347. The authors should remove 'Fig. 4', because of unnecessary.

12) L. 347. I think that descriptions of 29.57 cm and 34.83 cm is reversed.

13) L. 353-354. The authors should add descriptions for ', although it was not significantly different between CC and CS' in this sentence.

14) L. 389. I think that 'both in the CC and CS conditions' is correct.

15) L. 408-413. Only the genotype numbers are given in this part. As well as the other descriptions, the authors should also indicate accession names here.

16) L. 415, and Fig. 3. The authors should replace this figure to high resolution picture.

17) L. 447, and Fig. 4. The authors should indicate positions of the three selected accessions in this figure.

18) L. 486. I think 'indicated' would be correct.

19) L. 506-525. You should indicate G number for each accession name.

20) L. 532-543. Two-dimensional electrophoresis would be very useful for isolating individual proteins associated with strong cold tolerance. A lot of proteomics reports have been published on the identification of stress-related proteins by using 2D electrophoresis.

21) L. 544, and Fig. 8A. The authors should indicate individual accession names and whether it was CC or CS at the top pf gel picture.

22) L. 553, and Fig. 9. Please unify the fonts in this figure with those in the other figures.

23) In this study, the authors evaluated cold tolerance in 204 mungbean accessions. If you have genome sequencing data and whole genome SNPs for these accessions, you could carry out GWAS, identify novel gene loci, and collect novel genes involved in the control of phenotypic differences for cold tolerance among the 204 germplasm accessions.

Reviewer 2 Report

General comments

 Manasa et al., conducted a study entitled “Cold tolerance mechanisms in mungbean (Vigna radiata L.) genotypes during germination stage”. The manuscript contains publishable data; however, the draft needs major revisions for more clarity and readability to warrant a publication. For instance, authors used term cold stress, chilling stress, low temperature, or extreme conditions interchangeability; however, there is difference which authors might consider defining in the introduction section and use only one term onward which authors think is more appropriate for this paper. For instance, cold stress could be chilling (0-20 C) or freezing (below 0 C) whereas the low temperature could be any temperature value below plant optimum temperature requirements? Authors should provide the base germination temperature for mungbean so that readers can get idea how the low temperature affects germination. It would have been better to include a few other temperature treatments such as 5C, 15C to check how 204 genotypes behaved during these temperatures. Therefore, the authors should include reasoning why they selected only 10 C for this study and why not 5C or 15C? The figures are not very clear and some figures are blurred (PCA figure). Also, authors did 2 sets of experiment whereas presented only 30 values for 3 replications, either authors pooled data set or should highlight the total number of values for each parameter in each figure. For instance, each mean shows 30 or 60 values for 3 replications (1 set of experiments) or 6 replications (2 set of experiments) for more clarity. The authors also need to check the typos and grammatical errors for better clarity of the ideas and readability and understating of readers. For instance, authors mentioned name as Mansa versus Manasa in text and references list? The paper needs major revisions before it is accepted for publication in Agriculture-MDPI. See specific comments as follow.

Specific comments:

L2-3: Authors might consider deleting “stage” from the title.

L2 versus L12 versus L29: For consistency use same scientific term for mungbean. Apply same comment wherever applicable throughout the manuscript. See L67 for greenbean, L81 for mung, L83 for munggram as well.

L12-13: Sentence is not clear and missing words. Authors might consider rephrasing for more clarity.

L13-114: It is not clear from the sentence that low temperature is concern during only germination or it is also more important during vegetative and reproductive stage? Authors might consider rephrasing the sentence for more clarity considering title of the manuscript.

L13: “…………. yield losses…”?

L15: Missing word “…….. reduced grain filling?”, See flower drop, impairs anthesis??

L16-28: Authors might consider rearranging this section to present the results. Please note title states germination so present germination results first. It is also important to show here how many hours/days/weeks experiment was conducted and experimental treatments. Define acronyms first such as MDA, TTL1, MALDI-TOF MS? What does authors mean by “….. other germination parameters in L22”, better to include names for more clarity?

L31: Better to reorganize keywords alphabetically.

L34: Authors presented the idea of global warming, it would be better to write considering manuscript title. For instance, “………….climate change – clod stress – crop establishment – germination, etc.?”

L36: fulfil or fulfill?

L39: Authors need to pay special attention for typos throughout the manuscript. For instance, Mansa et al., 2021 or Manasa et al., 2021? Read manuscript carefully to address all typos and grammatical errors for better clarity and readability.

L39: “………with a very few…”?

L40-41: Better to differentiate cold temperature, chilling temperature, low temperature, etc. Authors might consider reading a review article entitled “Recent insights into cell response to cold stress in plants: Signaling, defense, and potential function of phosphatidic acid”, published by Wu et al., 2022.

L42: Cold stress or chilling stress? Authors might consider appropriate term according to the idea being presenting in each sentence.

L43-44: Provide reference.

L45: Authors might consider including recent references. Current reference is more than a decade old. See paper suggested above for recent references.

L45-50: Authors might consider all second messengers including phosphatidic acid.

L61-64: A very vague statement, authors might consider rephrasing this statement by providing time scale rather than “more rapidly faster” terms.

L64-66: Check both sentence and better present one idea versus two, could be harvested early versus longer period (for losing quality)?

L68: Include reference.

L73-74: Sentence is incomplete, for instance “…. no flowers and fruit loss?”

L75: “…. infew..”?

L74-75: Authors might consider proving base germination temperature for mungbean.

L79-81: chilling stress? “… decrease on mungbean productivity versus …… decrease in mungbean productivity”?

L85: term “several” is very vague, better to include names.

L87-88: Sentence is not clear to this reader especially “… its..”, either authors talking about a genotype versus several genotypes, or germination stage?

L83-88: Authors might consider including study hypothesis and specific research question(s) which they have addressed through this study.

L88: “……. extreme conditions.” is not clear to this reader. Do authors means low temperature or freezing temperature? Better to write with more clarity.

L91-93: Opening sentence states “A total of 200 genotypes whereas abstract says 204 versus sum of 127+58+15+4 = 204?

L93: Better to write complete title of Table 1 in supplementary file. For instance, “used in this study versus short title of study, provide complete names of genotypes in the table, better to add a column mentioning source of the genotypes”. 

L121: It is not clear to this reader that authors conducted 2 set of experiments whereas the data shown represent 30 individual values for 3 replications each of 10 seeds/seedlings. Either authors pooled data set for two separate experiments and presented as one?

L122-125: Did authors counted non-germinated seeds while counting germinated ones? There might be some seeds germinated but not emerged?

L165-166: Authors might consider providing some details how they selected 3 genotypes from 204. Also, were not 4 genotypes marked as check/control?

L176: 3ml versus 3 mL?

L312-313: Sentence is not clear to this reader. Grouped what? Also, temperature 22C is not cold stress?

L312-316: Here authors need more clarity while speaking regarding control. Either authors considering 22C as control or 4 genotypes (C1-C4).

L321: “….. thev…?”

L352: Drastic is a vague term, better to use significant in case the effects were significant.

Figure 3: This reader is unable to see figure 3 as it is very blur. Authors might also consider providing KMO value for PCA in the results.   

L570-571: Is it not a different statement than main objectives? Authors mentioned to establish a method for screening mungbean genotypes?

L666: Better to write only regarding plants instead of organisms.

L667: Avoid references in summary section.

L667: “The present study an approach…”?

L670: Authors should avoid using term sustainable yield as they have not measured it in current study.

L671: “… morphological more advantage……...”?

L672-673: TTLI is novel protein versus what authors mentioned in L631 or L673-674?

L664-677 versus L83: Main objective was to screen different genotypes for cold tolerance, so this reviewer suggests to consider rewriting the summary section to outline i. what parameters were used in cold stress screening, ii. Which genotypes showed more tolerance in terms of cold stress tolerance compared to what, what are the main reasons of such cold stress tolerance in authors’ point of view, and finally iii. What are the implications of such cold stress tolerance in terms of seedling establishment.

Reviewer 3 Report

In this study, the authors screened mung bean for cold tolerance-related genes at the germination stage to identify a new protein identified as TPR repeat containing thioredoxin TTL1 which highly correlated with the cold stress response of the genotypes. However, the entire manuscript needs revision before publication. The detail is followed:

Introduction

Line 83-88: “The present study… during germination stage.” “It also includes… susceptible germplasms.” “The primary step… under extreme conditions.” Reconsidering these three sentences makes the purpose of the experiment more obvious and logical.

Materials and Methods

Several varieties were selected for the experiment, and four high yielding released varieties were chosen as controls, whether the experimental data would be inaccurate due to the germination stage differences of the varieties themselves.

Results

3.1 Please adjust the figures to make the font uniform and label figure B and C.

Discussion

Line 633-663: “Cold stress induces… the respiration rate.” This section can be modified and placed in the introduction section. 

Round 2

Reviewer 1 Report

Thank you for making revision of your manuscript. I agree with your all corrections in this manuscript and your comments to my additional experimental suggestions.

Author Response

Answer to Reviewer/Editors comments:

  1. The manuscript has been verified. All relevant references have been included in the manuscript.
  2. The entire manuscript has also been edited.

Reviewer 3 Report

Thank you for your responses and revisions. However, there are some minor issues that need to be revised, the comments are as follows.

Materials and Methods

2.6 Integrate the experimental methods in this section so that the headings are numbered correctly

Thank you for your responses and revisions. However, there are some minor issues that need to be revised, the comments are as follows.

Materials and Methods

2.6 Integrate the experimental methods in this section so that the headings are numbered correctly

Author Response

Answer to Reviewer Comments:

  1. As per the reviewer comments, the manuscript has been edited and corrected.
  2. As per the suggestion, the numbering if the material & method section  has been corrected
  3. All the references have also been verified and corrected.
